The tetrapod fauna of the upper Permian Naobaogou Formation of China: 5. Caodeyao liuyufengi gen. et sp. nov., a new peculiar therocephalian

Liu Jun liujun@ivpp.ac.cn 1 2 3
Abdala Fernando nestor.abdala@wits.ac.za 4 5
1 Key Laboratory of Vertebrate Evolution and Human Origins of Chinese Academy of Sciences, Institute of Vertebrate Paleontology and Paleoanthropology, Chinese Academy of Sciences , Beijing , China
2 College of Earth and Planetary Sciences, University of Chinese Academy of Sciences , Beijing , China
3 CAS Center for Excellence in Life and Paleoenvironment , Beijing , China
4 Evolutionary Studies Institute, University of the Witwatersrand , Johannesburg , South Africa
5 Unidad Ejecutora Lillo (CONICET-Fundación Miguel Lillo) , Tucuman , Argentina
Knoll Fabien
Electronic publication date: 2020 May 28
Publication date: 2020
Volume: 8
Electronic Location ID: e9160
Received 2020 Feb 17; Accepted 2020 Apr 18
Copyright: ©2020 Liu and Abdala
Copyright year: 2020
Copyright holder: Liu and Abdala
License: This is an open access article distributed under the terms of the Creative Commons Attribution License, which permits unrestricted use, distribution, reproduction and adaptation in any medium and for any purpose provided that it is properly attributed. For attribution, the original author(s), title, publication source (PeerJ) and either DOI or URL of the article must be cited.
License URL: https://creativecommons.org/licenses/by/4.0/

Keywords: Therocephalia, Upper Permian, Naobaogou Formation

Funding: Strategic Priority Research Program of Chinese Academy of Sciences XDB26000000 International Partnership Program of Chinese Academy of Sciences 132311KYSB20190010 National Natural Science Foundation of China 41572019 Chinese Academy of Sciences President’s International Fellowship Initiative Grant 2016VBB054 Jun Liu was supported by the Strategic Priority Research Program of Chinese Academy of Sciences (No. XDB26000000), the International Partnership Program of Chinese Academy of Sciences (No. 132311KYSB20190010), and the National Natural Science Foundation of China (No. 41572019). Fernando Abdala was supported by the Conicet of Argentina, NRF of South Africa, and a research trip to China funded by the Chinese Academy of Sciences President’s International Fellowship Initiative Grant (2016VBB054). The funders had no role in study design, data collection and analysis, decision to publish, or preparation of the manuscript.

==============================
The upper Permian Naobaogou Formation has been the goal of recent contributions that notably increased the knowledge of its terrestrial vertebrate fauna and unravelled a hidden late Permian therocephalian diversity in China. Two very different species of therocephalians have been documented in the Naobaogou Formation and they were recovered as basal akidnognathids in cladistic analyses. In this contribution we describe Caodeyao liuyufengi gen. et sp. nov., represented by a partial skull and mandible, and a humerus. The new taxon features a short, high snout and a wide temporal opening with the coronoid process of the mandible separated by a wide space medially to the zygomatic arch. The latter feature is only recognized in the Russian therocephalian Purlovia maxima and it is also characteristic of non-mammaliaform cynodonts. Phylogenetic analysis indicates a close relationship of the new Chinese taxon with Purlovia maxima, producing a monophyletic Laurasian group in therocephalian phylogeny. With the representation of three different species, the Naobaogou Formation is now the most prolific unit documenting therocephalian late Permian diversity in China.

Introduction

Chinese tetrapods of late Permian age are well known in Xinjiang and North China (Li, Wu & Zhang, 2008; Liu, 2018). Recent systematic explorations in exposures of the Naobaogou Formation in the Daqingshan area have resulted in a significant expansion of the fossil tetrapod record. The fauna of this unit was originally represented by the dicynodont Daqingshanodon limbus (Zhu, 1989), with the later addition of the captorhinid Gansurhinus qingtoushanensis (Li & Cheng, 1997; Reisz et al., 2011). Recent contributions added two new akidnognathid therocephalians, Shiguaignathus wangi and Jiufengia jiai (Liu & Abdala, 2017a; Liu & Abdala, 2019), the pareiasaur Elginia wuyongae (Liu & Bever Gabriel, 2018), and several additional dicynodonts (Liu, 2019).

The Chinese therocephalian record thus far consists solely of eutherocephalians, with nearly all of the extensive Triassic record of this group pertaining to the subclade Baurioidea. In the Permian, two of the three known taxa (Jiufengia and Shiguaignathus) belong to the majority-Gondwanan clade Akidnognathidae. The phylogenetic placements of two Chinese therocephalians, the late Permian Dalongkoua fuae (Liu & Abdala, 2017b) and the Triassic Yikezhaogia megafenestrala (Li, 1984), are uncertain, as these species have not yet been included in a phylogenetic analysis.

The recent discoveries of two new therocephalians from the Naobaogou Formation (Liu & Abdala, 2017a; Liu & Abdala, 2019) provided the first records of this therapsid lineage in the unit, and represent only the second and third therocephalians known from the late Permian of China. Here we report an additional therocephalian taxon from the Naobaogou Formation, Caodeyao liuyufengi gen. et sp. nov., with peculiar cranial features resembling the enigmatic Russian taxon Purlovia maxima. This contribution is part of a series documenting therocephalian diversity in the late Permian of China, which shows that the evolution of this group in Laurasia was more complex than previously known.

Nomenclatural acts

The electronic version of this article in portable document format will represent a published work according to the International Commission on Zoological Nomenclature (ICZN), and hence the new names contained in the electronic version are effectively published under that Code from the electronic edition alone. This published work and the nomenclatural acts it contains have been registered in ZooBank, the online registration system for the ICZN. The ZooBank Life Science Identifiers (LSIDs) can be resolved and the associated information viewed through any standard web browser by appending the LSID to the prefix http://zoobank.org/. The LSID for this publication is: urn:lsid:zoobank.org:pub:EBA6DC8E-45B6-42C7-B570-BFA9559F2795. The online version of this work is archived and available from the following digital repositories: PeerJ, PubMed Central, and CLOCKSS.

Systematic Paleontology

THERAPSIDA Broom (1905)	
THEROCEPHALIA Broom (1903)	
EUTHEROCEPHALIA Hopson & Barghusen (1986)	
Caodeyao liuyufengi gen. et sp. nov.	

Etymology. ‘Caodeyao’ refers to the village where the fossil was collected; the species name is dedicated to Mr. Liu Yu-Feng, the technician and driver who made significant contributions to the Daqingshan field trips.

Holotype. IVPP V 23298, partial skull with mandible, left humerus.

Type Locality and Horizon. Locality DQS42 (N40°42′39″, E110°36′56″), Caodeyao village, Tumed Right Banner, Nei Mongol, China; middle part of Member II, Naobaogou Formation.

Diagnosis. A small, short-snouted eutherocephalian differentiated from all taxa except Purlovia maxima by a deep, laterally-bowed zygomatic arch. It is distinguished from P. maxima by having a tall pterygoid median crest, a wide, flat base of the postorbital bar, six upper postcanines, and small denticles on pterygoid transverse process. The following characters may represent autapomorphies of Caodeyao liuyufengi (though this is uncertain given the limited preservation of the braincase and postcranium in many therocephalians): very large fenestra ovalis and deltopectoral crest longer than half the length of the humerus.

Description

As a result of erosion to the skull roof, the palate is exposed in both dorsal and ventral views (Figs. 1–4). The snout is deflected to the left side and the braincase is broken into two parts along a wide crack (Figs. 1 and 4). The mandibles tightly occlude with the skull, and the posterior portion of the right ramus is missing. A left humerus was found associated with the cranial material.

Figure 1 Holotype of Caodeyao liuyufengi (IVPP V 23298) in dorsal view: (A) photo and (B) drawing of skull with mandibles; (C) photo and (D) drawing of occiput and braincase.

Abbreviations: Ar, articular; avSq, anteroventral process of squamosal; ch, choana; cp, coronoid process; D, dentary; Ec, ectopterygoid; Ep, epipterygoid; fm, foramen magnum; J, jugal; L, lacrimal; lc, lower canine; lf, lacrimal foramen; lpr, lacrimo-palatine ridge; M, maxilla; mf, mandibular fenestra; Par, Prearticlar; Pf, prefrontal; pfOp, paroccipital fossa of opisthotic; Pl, palatine; Pm, premaxilla; Pr, prootic; pr, parasagittal ridge; ps, palatine sinus; Pt, pterygoid; ptf, post-temporal fenestra; Q, quadrate; Qj, quadrtojugal; qrPt, quadrate process of pterygoid; Sa, surangular; sH, sinus Highmori; So, supraoccipital; sov, suborbital vacuity; Sq, squamosal; tp, transverse process of pterygoid; uc, upper canine; V, vomer. Photo credit: Wei Gao. Drawing credit: Jun Liu.

Figure 2 Holotype of Caodeyao liuyufengi (IVPP V 23298), skull with mandibles: (A) photo and (B) drawing in left view, (C) photo in right lateral view.

Abbreviations: A, angular; D, dentary; J, jugal; L, lacrimal; lc, lower canine; lui, last upper incisor; M, maxilla; mf, mandibular fenestra; Pf, prefrontal; Po, postorbital; Q, quadrate; Qj, quadrtojugal; Sa, surangular; Sp, splenial; Sq, squamosal; tp, transverse process of pterygoid; uc, upper canine; ui, upper incisor. Depressions showed in light cyan. Photo credit: Wei Gao. Drawing credit: Jun Liu.

Figure 3 Holotype of Caodeyao liuyufengi (IVPP V 23298), skull with mandibles: (A) photo and (B) drawing in dorsomedial view.

Abbreviations: A, angular; Ar, articular; cb, canine boss; cp, coronoid process; Cr, coronoid; D, dentary; Ec, ectopterygoid; J, jugal; L, lacrimal; lc, lower canine; ler, lacrimo-ectopterygoid ridge; lpr, lacrimo-palatine ridge; M, maxilla; mf, mandibular fenestra; Par, Prearticlar; Pf, prefrontal; Pl, palatine; Po, postorbital; ps, palatine sinus; Pt, pterygoid; Q, quadrate; Qj, quadrtojugal; Sa, surangular; sH, sinus Highmori; sov, suborbital vacuity; Sq, squamosal; tp, transverse process of pterygoid; uc, upper canine; V, vomer. Photo credit: Wei Gao. Drawing credit: Jun Liu.

Figure 4 Holotype of Caodeyao liuyufengi (IVPP V 23298) in ventral view: (A) photo and (B) drawing of skull with mandibles; (C) photo and (D) drawing of occiput and braincase.

Abbreviations: A, angular; Ar, articular; avSq, anteroventral process of squamosal; Bo, basioccipital; Cr, coronoid; crch, crista choanalis; D, dentary; Ec, ectopterygoid; fm, foramen magnum; fo, fenestra ovalis; icf, internal carotid foramen; imc, intermediate crest of pterygoid; ipOp, internal process of opisthotic; jf, jugular foramen; M, maxilla; mc, pterygoid median crest; Par, Prearticlar; Pl, palatine; Ps, parabasispenoid; Pt, pterygoid; Q, quadrate; Qj, quadrtojugal; qrPt, quadrate process of pterygoid; Sp, splenial; Sq, squamosal; tp, transverse process of pterygoid; tso, spheno-occipital tubercle; V, vomer. Photo credit: Wei Gao. Drawing credit: Jun Liu.

Skull

The incomplete skull measures nearly 13 cm along the midline, and the estimated total skull length is ∼14 cm. The snout is short, representing approximately 40% of the skull length. A small portion of the premaxilla is preserved with the last left incisor (Fig. 1), and it is partially covered by the maxilla laterally. A diastema the length of two incisors is present between the last incisor and the upper canine. The ventral margin of the maxilla slopes upwards anterior to the canine but is nearly straight in the postcanine region (Fig. 2). The preserved portion of the maxilla suggests that the facial plate is relatively high, its height roughly equal to half of its anteroposterior length. In dorsal view, the snout is constricted behind the canine on the better-preserved left side (Fig. 1) (the right side is incomplete, with the canine root exposed). The maxilla bears numerous pits, depressions and foramina on its external surface (Fig. 2). A depression extends anteroventrally between the canine buttress and the orbit. The right maxilla houses the crowns of one canine and six postcanines, whereas the left preserves only the canine and last postcanine. The postcanines are conical with a smooth surface. A lengthy diastema is present between the canine and the first maxillary postcanine.

Erosion of the skull roof has exposed the inner surface of the snout (Figs. 1 and 3). The canine boss is transversely narrower than that of Shiguaignathus wangi (Liu & Abdala, 2017a; Liu & Abdala, 2017b). The antrum Highmori is not preserved, and the sinus Highmori is almost triangular in dorsal and medial views. The lacrimo-palatine ridge is nearly vertical and very deep (Figs. 1 and 3). The palatine sinus can be divided into two parts: a lateral part with a flat floor and a medial part forming a pocket. The medial margin of the palatine sinus is a slender parasagittal ridge on the palatine (Figs. 1 and 3).

Only the ventral portion of the left orbit is preserved (Figs. 1–3). The anterior border lies within the anterior half of the skull, an indication of the proportionally short snout. The preserved orbital margin is formed by the prefrontal (preserved only in its ventral portion), lacrimal, and jugal (Fig. 2). The lacrimal is a nearly triangular bone in lateral view, having a short dorsal suture with the prefrontal and forming a thickened ridge on the anterior and ventral margins of the orbit. It bears a pit, the lacrimal foramen, within the orbit wall (Fig. 1). The anterior extension of the jugal is restricted to the middle portion of the orbit; so the deep suborbital bar is formed by the maxilla and lacrimal anteriorly, the lacrimal and jugal towards the center and somewhat posteriorly, and only the jugal on the posterior portion (Figs. 2 and 3). The anterior process of jugal bears a distinct depression on the lateral surface, just posterior to the suture with the maxilla. The jugal has a robust dorsal process, encircling the orbit posteriorly, though the dorsal tip is missing where it contributes to the postorbital bar. The jugal extends posteriorly towards the occiput, and is overlapped laterally by the tall, triangular anterior process of the squamosal. The deep subtemporal zygomatic arch is formed by the jugal and squamosal. Behind the orbit, the zygomatic arch bows laterally, and has a concave ventral margin throughout its length. The temporal fenestra is widest toward its posterior margin (Fig. 1).

The squamosal forms most of the posterolateral portion of the temporal region (Figs. 1 and 2), but the dorsal portion of the bone is missing so its full extent is uncertain. Its posteroventral process bears a notch housing the dorsal portion of the quadrate-quadratojugal complex, which is exposed in lateral view within the ‘squamosal slit’ (Fig. 2). Medioventrally, the squamosal broadly contacts the paroccipital process of the opisthotic (Figs. 4, 5C, 5D). Anteroventrally, it bears a long process, which probably contacted the prootic, epipterygoid, and quadrate process of the pterygoid as in other eutherocephalians. However, only the contact with the pterygoid is preserved. The squamosal has an anteriorly directed, triangular lamina which roofs the post-temporal fenestra (Figs. 5E, 5F), but it is separated from the border of this fenestra by a ventral extension of the tabular in occipital view. A broad, shallow sulcus lies on the posterior surface of the squamosal just lateral to the paroccipital process of the opisthotic (Figs. 5C, 5D).

Figure 5 Holotype of Caodeyao liuyufengi (IVPP V 23298): incomplete skull, occiput and braincase.

Incomplete skull: (A) photo and (B) drawing in posterior view. Occiput: (C) photo and (D) drawing in posterior view, (E) photo and (F) drawing in anterior view. Braincase: (G) photo and (H) drawing in anteromedial view from right side, (I) photo and (J) drawing in dorsolateral view from left side, (K) photo and (L) drawing in ventrolateral views from left side, and (M) photo in dorsomedial view. Abbreviations: adp, anterodorsal process of prootic; Ar, articular; avp, anteroventral process of prootic; Bo, basioccipital; cc, crus communis canalium; cEp, contact to epipterygoid; D, dentary; Eo, exoccipital; Ep, epipterygoid; flf, floccular fossa; fm, foramen magnum; fo, fenestra ovalis; hsc, horizontal semicircular canal; ipr, incisura prootica; J, jugal; jf, jugular foramen; L, lacrimal; Op, opisthotic; pdpOp, posterodorsal process of opisthotic; pdpPr, posterodorsal process of prootic; Po, postorbital; pop, paraoccipital process; Pr, prootic; Ps, parabasispenoid; ptf, post-temporal fenestra; pvs, posterior vertical semicircular canal; Q, quadrate; Qj, quadrtojugal; qrOp, quadrate process of opisthotic; qrPt, quadrate process of pterygoid; Sa, surangular; scr, sacculo-cochlear recess; So, supraoccipital; Sq, squamosal; st, sinus tubera; T, tabular; tso, spheno-occipital tubercle; vii, foramen for nerve VII. Scale bars equal 1 cm. Photo credit: Jun Liu and Wei Gao. Drawing credit: Jun Liu.

The vomer is an elongated element with a long and very narrow interchoanal process, resembling that of Kotelcephalon viatkensis (Tatarinov, 1999). Although the anterior tip of the interchoanal process is missing, the putative contact of the anterior process with the premaxilla appears to be much narrower than in most therocephalians. The interchoanal portion of the vomer generally has a smooth ventral surface, although its posterior tip (near the margin of the internal choana) bears a low ventromedian crest continuing on to its palatal surface. The dorsal surface of the vomer is incompletely preserved, but three close ridges can be observed on the dorsal side, with a narrow space between them (Figs. 1 and 3). Dorsally, the vomer rises as a triangular flange near the posterior margin of the choana (Figs. 1 and 3). The posteriormost surface of the vomer, where it contacts the palatines and forms part of the main palatal surface, is fairly flat. Dorsally, the posterior part of the vomer and the pterygoid form a midline ridge. This ridge diminishes at the level of the transverse process of the pterygoid (Fig. 1).

The palatines are elongate and broad on the palate, and separated medially by the vomer and the pterygoids. On the dorsal side, the palatines extend near the midline of the skull, but are still separated by a narrow section of vomer (Fig. 1). In ventral view, the palatines border the posterolateral margin of the choana, with the maxilla forming the anterolateral margin. The crista choanalis extends medially from the canine boss to the anterior corner of the suborbital vacuity (Fig. 4). The middle section of the crista choanalis is low and indistinct.

The large pterygoid bears the palatal process, transverse process, and posteriorly, the diverging quadrate ramus (Figs. 1, 3 and 4). The palatal process bears a tall median crest, much more developed than the paired intermediate crests lateral to it. There is no interpterygoid vacuity. The ventrally-directed transverse process expands ventrolaterally, terminating in a blunt lateral tip. The anterior face of the transverse process bears a broad, shallow depression. The posteromedial portion of the transverse process is dentigerous. The process posteriorly abuts a well-developed, subtriangular suborbital vacuity with a nearly straight medial margin. It is bordered by the palatine medially, the palatine and ectopterygoid laterally, and the pterygoid posteriorly.

A small ectopterygoid forms part of the transverse process (Figs. 1, 3 and 4). This bone forms the anteromedial border of the subtemporal fossa. The major part of the ectopterygoid is curved, forming a nearly triangular lamina with the transverse flange of the pterygoid. The lateral side of this lamina forms a buttress pressed against the mandible.

The dorsal portion of the occiput and the portion anterior to the foramen magnum are not preserved, and a portion of the inner side of the braincase is exposed (Figs. 1 and 5). There is an enormous fenestra ovalis, exposed in ventral, lateral and medial views, with a diameter of nearly 10 mm (Figs. 5G–5M). It is formed by the prootic dorsally, the opisthotic posteriorly, and the parabasisphenoid and part of the basioccipital ventrally. The prootic is incomplete anterior to the foramen for cranial nerve VII, and its anteroventral process is also incomplete. On its dorsal margin, the prootic incisura is shallow, and the anterodorsal process is short (Figs. 5I, 5J). The ventral side of the prootic is raised as a thickened ridge on the anterior surface of the occiput, bordering the fenestra ovalis and contacting the parabasisphenoid. Posterodorsally, the prootic contacts the supraoccipital. On the medial surface, where the prootic meets the supraoccipital, lies a wide, shallow, conical floccular (subarcuate) fossa for the lateral lobe of the cerebellum (Fourie, 1974) (Figs. 5G, 5J). Posteroventral to this fossa is the crus communis canalium, which represents the dorsal limit of the sinus utricularis. This recess extends dorsally and anteriorly to meet the floccular fossa. Posteriorly, a shallow recess for the posterior vertical semicircular canal runs vertical to the jugular foramen. The canals and fossae of this region are similar to those of Russian therocephalians Chthonosaurus velocidens and Karenites ornamentatus (Ivakhnenko, 2011), but the fenestra ovalis is much larger in Caodeyao. The prootic is sutured with the opisthotic posteriorly and it has an incompletely preserved posteroventral process. The slender posterodorsal process of the prootic is not exposed on the anterior side, only on the occiput (Fig. 5C, 5D). There is no evidence of the presence of a central process of the prootic. The epipterygoid is partially preserved, with the base of its ascending process present above the quadrate process of the pterygoid (Figs. 1 and Figs. 5A, 5B).

The opisthotic is a stout bone with a wide, long posterodorsal process, a slender internal process, and a stout paroccipital process. The posterodorsal process, medial to the post-temporal fenestra, contacts the tabular laterally and the supraoccipital dorsally (Figs. 5C, 5D). The internal process forms a thin lamina between the fenestra ovalis and the jugular foramen (Figs. 4 and 5M). In ventral view on the left side, this process is exposed as a ridge (Fig. 4). The paroccipital process can be divided into an anteroventral flange and a posterodorsal flange, which join to form a V-shaped cavity, the paroccipital fossa of the opisthotic (Fig. 1). The anteroventral flange contacts the posteroventral process of the prootic anteriorly, and their dorsal surface forms the floor of the post-temporal fenestra (Figs. 5C–5F). The lateral side of the paroccipital process is subdivided into a mastoid process on the occipital side and a quadrate process on the palatal side, the latter more massive than the former (Fig. 4). The sacculo-cochlear recess is located anteromedial to the internal process of the opisthotic and medial to the fenestra ovalis (Fig. 5M).

The large, elliptical post-temporal fenestra is positioned at the same level as the foramen magnum. It is bordered by the paroccipital process ventrally, the posterodorsal process of the prootic medially, and the tabular and squamosal dorsally and laterally. On the occiput, the tabular, rather than the squamosal, forms the margin of post-temporal fenestra (Figs. 5E, 5F), as in the Russian Viatkosuchus and Annatherapsidus (Ivakhnenko, 2011), but differing from the South African Moschorhinus and Theriognathus (Durand, 1991; Huttenlocker & Abdala, 2016).

In occipital view, the round foramen magnum is formed by the supraoccipital dorsally, the exoccipital laterally, and the basioccipital ventrally (Figs. 5C, 5D). The supraoccipital contacts the tabular laterally, the opisthotic ventrolaterally, and the exoccipital ventrally. Its dorsal margin is not preserved. The paired exoccipitals do not meet in the midline. In occipital view, the exoccipital can be divided into a flat triangular dorsal part and a ventral projection that should form part of the occipital condyle. The posterior part of the basioccipital, which should form the median portion of the tripartite occipital condyle, is missing.

The parabasisphenoid complex is broken and the middle part is missing (Figs. 4 and 5). Anteriorly, it is buttressed against the pterygoid posterior to the medial portion of the quadrate process. An interdigitated suture is present between these two elements, and near the suture are the paired internal carotid foramina. A parasagittal fossa extends posterior to the opening of the internal carotid foramen on the ventral surface of the parabasisphenoid. No cultriform process is observed. The anteriormost portion of the parabasisphenoid features a short and low median keel, with the anterior portion of the keel bifurcated into two ridges. These ridges continue anteriorly onto the pterygoid, with a concave area between them. Posteriorly, the basisphenoid flares to form the deep spheno-occipital tubercles with the basioccipital.

The left quadrate and quadratojugal are displaced dorsolaterally, suggesting a loose articulation with the squamosal (Figs. 1–5). The quadratojugal is a slender rod tightly sutured to the quadrate (Fig. 1). These bones are separated by a quadrate-quadratojugal foramen. The quadrate is composed of the dorsal process and the ventral transversely-elongated trochlea (Figs. 3, 5A). In anterior view, the dorsal process has a concave depression with lateral and medial crests. The medial crest is partially exposed as a flat surface that probably contacted the paroccipital process (Fig. 3). Only the lateral condyle of the quadrate trochlea is exposed (Figs. 2, 5A). Dorsolaterally, the trochlea supports the base of the quadratojugal.

Mandible

The left mandible and anterior portion of the right ramus are occluded to the skull. The left mandible measures 126 mm in length. The mandibular rami diverge posterolaterally with an angle of 45° in ventral view.

The dentary is a robust bone with a stout, vertical symphysis forming a chin (Fig. 2). A robust, vertical anterior symphyseal region is known in several therocephalian groups including basal therocephalians such as Gorynychus, Lycosuchus, and some scylacosaurids (Abdala et al., 2014b; Abdala, Rubidge & Van den Heever, 2008; Kammerer & Masyutin, 2018; Pusch et al., 2020). This condition is also present in some eutherocephalians, including akidnognathids such as Moschorhinus, Olivierosuchus, and Promoschorhynchus (“mental protuberance”) (Botha-Brink & Modesto, 2011; Durand, 1991; Huttenlocker, Sidor & Smith, 2011) and bauriamorphs such as Nothogomphodon (Ivakhnenko, 2011; Liu & Abdala, 2015), and Microgomphodon (Abdala et al., 2014a). Development of a tall mandibular symphysis is extreme in the Russian Purlovia, in which there is a marked ventral chin (Ivakhnenko, 2011: Fig. 8).

The dentary height decreases posteriorly and reaches its shallowest point near the dentary mid-length, where it is slightly constricted such that the dentary is somewhat concave ventrally. The lateral dentary surface bears a well-developed sulcus immediately behind the symphysis, ending at the posterior margin of the bone, above the angle of the dentary (Fig. 2). The angle of the dentary, formed by the confluence of the posteroventral margin of the horizontal ramus and the anteroventral margin of the ascending ramus, is about 140°. The coronoid process is well developed, with a wide lateral surface, and its dorsal margin ends slightly above the level of the base of the orbit. Posterodorsally, the coronoid process slightly expands in width and its posterodorsal margin is curved dorsally and medially (Fig. 3). The posterior margin of the dentary borders a dorsoventrally narrow, elongate mandibular fenestra. Four incisors appear to be present on the right side of the mandible, but one of them may prove to be an upper tooth. The left dentary bears one canine and six postcanines.

The splenial is a long bone which covers the lower half of the dentary medial surface and does not reach the lower margin of the dentary (Figs. 2 and 4). The splenials from both sides meet on the posteroventral surface of the symphysis.

Only the posterodorsal corner of the coronoid bone is exposed, the rest being covered by the transverse flange of the pterygoid (Fig. 3). It extends ventrally to partially cover the anterior process of the prearticular.

The thin prearticular extends anteriorly into the medial trough of the dentary, runs along the medial side of the angular, and ends posteriorly to contact the articular, but sutures with the latter are not visible. (Figs. 3 and 4)

The angular is located medial and posterior to the dentary (Fig. 2). It forms most the lateral surface of the postdentary area. Its ventral margin is located dorsal to that of the dentary. It borders the narrow mandibular fenestra ventrally. The reflected lamina is elongated posteriorly with a spade-like shape. It is ornamented with fine, radiating ridges and grooves and bears a deep dorsal notch that is positioned at the level of the posterior margin of the coronoid process of the dentary. The ventral margin of the reflected lamina is at the same level as that of the dentary. The angular is bordered dorsally by the surangular.

The surangular is exposed posteriorly in the convex mandibular margin dorsal to the angular in lateral view, with the major portion of the bone covered by the dentary laterally (Figs. 2 and 3). It forms the dorsal rim of the mandibular fenestra on the medial side of the dentary. Anteriorly, it contacts the coronoid and the prearticular. An irregular fossa is formed medial to the dentary, between the surangular and the coronoid, behind the transverse process. A shallow fossa lies on the medial surface of the surangular, dorsal to the posterior end of the prearticular, possibly for the insertion of a portion of the pterygoideus musculature (Fig. 3).

The articular lies on the posterior end of the lower jaw and forms the jaw joint with the quadrate-quadratojugal complex. This complex is displaced from its original position and thus the articular condylar surface is exposed. It bears two concave facets for the quadrate condyles and a wide medial triangular flange.

Humerus

The left humerus is the only preserved postcranial element (Fig. 6). It is a robust, elongated bone with slightly twisted proximal and distal ends. The proximal end has a proximodorsally bulbous head. The proximal half is curved dorsally relative to the distal half. The keel-like deltopectoral crest is well developed, extending down more than half the length of the bone. The distal end is incomplete and deformed with an unnatural strong curvature. A narrow fossa is developed on the dorsal surface of the mid-shaft. A prominent entepicondylar foramen is present.

Figure 6 Holotype of Caodeyao liuyufengi. (IVPP V 23298), left humerus in (A) ventral, (B) anterolateral, (C) dorsal, and (D) posteromedial views.

Abbreviations: bi gr, bicipital groove; cp, capitulum; d c, deltopectoral crest; f br, fossa related to brachialis origin; f en, entepicondyle foramen; g t, greater tuberosity; h, humeral head; l t, lesser tuberosity; th, trochlea. Photo credit: Jun Liu.

Discussion

Credible records of late Permian therocephalians in China have only recently been confirmed (Liu & Abdala, 2017a; Liu & Abdala, 2017b; Liu & Abdala, 2019). Two of these findings are basal akidnognathids: Shiguaignathus wangi and Jiufengia jiai from the Naobaogou Formation (Liu & Abdala, 2017b; Liu & Abdala, 2019). The current report of Caodeyao liuyufengi, the first non-akidnognathid therocephalian from this fauna, demonstrates increased taxonomic diversity in the Naobaogou Formation.

Caodeyao liuyufengi can be recognized as a therocephalian by the presence of large suborbital vacuity, the palatal fenestra confluent with the internal naris, and the splenial laterally obscured by the dentary (Hopson & Barghusen, 1986; Huttenlocker, 2009). It can be diagnosed as an eutherocephalian by the presence of a fused vomer, the deeply hollowed dorsal surface of the paroccipital process forming the floor of the post-temporal fenestra, a laterally visible mandibular fenestra, and the (putative) presence of four lower incisors. However, instead of a broad vomerine interchoanal process, as in most eutherocephalians including akidnognathids (Huttenlocker & Abdala, 2016; Ivakhnenko, 2011; Liu & Abdala, 2017a; Liu & Abdala, 2017b; Liu & Abdala, 2019), Caodeyao liuyufengi has a quite narrow interchoanal process, more similar to the non-eutherocephalians Gorynychus masyutinae (Kammerer & Masyutin, 2018) and Kotelcephalon viatkensis (Tatarinov, 1999; Ivakhnenko, 2011).

Caodeyao has a zygomatic arch that is moderately deep and bowed laterally, producing a wide space separating it from the lateral surface of the mandible. The zygomatic arch is slender in most therocephalians, with the exceptions of Lycosuchus (Abdala et al., 2014a; Abdala et al., 2014b: Fig. 1.1), Glanosuchus (SAM-PK-637), Traversodontoides (Young, 1974), and Purlovia (Ivakhnenko, 2011). A transversely expanded temporal fenestra is also present in whaitsiids, but in those taxa the mandible generally still contacts the edge of the zygomatic arch. The laterally bowing of the zygomatic arch and separation from the mandible in Caodeyao is more similar to the Russian Purlovia (Ivakhnenko, 2011) and several non-mammaliaform cynodonts, such as Procynosuchus and Thrinaxodon (Brink, 1963; Kemp, 1979; Hopson & Kitching, 2001; Jasinoski, Abdala & Fernandez, 2015). Unlike those cynodonts, however, the greatest width of the temporal opening is at its posterior edge.

Purlovia is a peculiar therocephalian, featuring an incipient maxillary platform lateral to the postcanine series (Fig. 7), also known in the whaitsiid Theriognathus and in some bauriids (Abdala et al., 2014a; Huttenlocker & Abdala, 2016). In addition, the facial portion of the maxilla external to the dentition in Purlovia is densely ornamented with low rounded tubercles (Ivakhnenko, 2011). Purlovia was initially referred to Nanictidopidae (Watson & Romer, 1956), a family that originally included several disparate therocephalians such as Choerosaurus, Hofmeyria, and Promoschorhynchus. However, the family Nanictidopidae sensu Ivakhnenko (2011) included only Nanictidops and Purlovia, based on the shared presence of a wide temporal region. In the brief, original description of Nanictidops kitchingi (Broom, 1940: 165–167), there is no report of diagnostic characters and the most remarkable features that can be mentioned are the short snout (Figs. 8A, 8B) and an “imperfect secondary palate” (Broom, 1940) formed by short overlap of the maxillae on the vomer (Figs. 8C, 8B). This palatal morphology is known in some non-bauriid baurioids such as Ictidosuchops (Hopson & Barghusen, 1986), although the short and wide snout and the enormous size of the canine and postcanine in RC 49 (holotype and only known specimen of Nanictidops; Fig. 8B) differentiate it from the delicate condition of the snout and dentition of Ictidosuchops. In general, there is little justification to consider the South African Nanictidops in the same group as the Russian Purlovia, something that is also supported by the results of our phylogenetic analysis (Fig. 9).

Figure 7 Holotype of Purlovia maxima (PIN 1538/47): palate and zygoma in (A) ventral and (B) dorsal views; (C) zygoma in lateral view; (D) right mandible in lateral view.

Abbreviations: An, angular; cb, canine boss; chi, mandibular chin; cp, coronoid process; crch, crista choanalis; De, dentary; fa, mandibular facet between the posterior tip of the dentary and the surangular (highlighted in transparent light grey); J, jugal; L, lacrimal; lpr, lacrimo-palatine ridge; M, maxilla; mp, maxillary platform; mt, maxillary tubercles; pc, postcanine; Pl, palatine; pob, postorbital bar; ps, palatine sinus; Sa, surangular; sH, sinus Highmori; Sq, squamosal; uc, upper canine; zy, zygoma. Scale bars equal 1 cm. Photo credit: Fernando Abdala.

Figure 8 Holotype of Nanictidops kitchingi (RC 49): skull in (A) dorsal and (B) right lateral views; section of skull (indicated in B) in (C) dorsal and (D) ventral views.

Abbreviations: ch, choana; cvm, contact between vomer and maxilla in the palate; F, frontal; i, upper incisor; M, maxilla; N, nasal; o, orbit; pc, postcanine; Pf, prefrontal; Pm, premaxilla; Po, postorbital; prc, precanine; sn, snout; sov, suborbital vacuity; te, temporal opening; uc, upper canine; V, vomer; zy, zygoma. Scale bars equal 2 cm. Photo credit: Fernando Abdala.

Figure 9 Major consensus tree of Therocephalia relationships. Caodeyao is indicated in red.

Numbers above the branch indicate Bremer support of the clades (only values of two or more are indicated), numbers below the branch indicate frequency of clades in the fundamental trees (frequency as 100% is not shown).

To test the phylogenetic position of Caodeyao, it was coded for our recent data matrix (Liu & Abdala, 2019) with the additions of Purlovia and Nanictidops (Appendix). The coding of the number of the lower incisors in Lycosuchus was changed from four to three, following a recent redescription of this taxon (Pusch et al., 2020). The matrix was analyzed with TNT 1.5 (Goloboff & Catalano, 2016). Gorgonopsia is used as the outgroup, the routine followed for the search of most parsimonious trees (mpt) consisted of 1,000 random addition sequences and TBR, saving 100 trees per replications, and a second search using the trees from RAM as starting point and implementing TBR on those trees. Seventeen multistate characters were treated as additive. The search resulted in 168 mpt of 400 steps in which the major groups of Therocephalia are recovered as monophyletic, including Scylacosauridae, Chthonosauridae, Akidnognathidae, Whaitsioidea, and Baurioidea, a result is nearly identical to Fig. 10 of Liu & Abdala (2019). In this analysis, the relationships of Gorynychus and Lycosuchus with Scylacosauridae are still unclear. Nanictidops is recovered as the basalmost member of Baurioidea in most trees, and this clade is supported by the maxillary palatal processes contacting or nearly contacting the ventrally extending vomer (39) and the presence of functional upper precanine maxillary teeth in adults (94). In a few trees, Nanictidops is recovered as the sister taxon of the unnamed clade X shown in Fig. 9.

The phylogenetic analysis indicates a sister-group relationship between Caodeyao and Purlovia, making up a therocephalian clade represented exclusively in Laurasia. This new clade is supported by a deep suborbital bar (11) and a moderately deep zygomatic arch (23). Although Caodeyao and Purlovia are represented by incomplete skulls, they can be easily differentiated from each other: there are six upper postcanines in Caodeyao vs. eight to nine in Purlovia; small denticles on the transverse process in Caodeyao, which are absent in Purlovia; the median crest of pterygoid is taller in Caodeyao than in Purlovia; and the base of the postorbital bar is wider and flatter in Caodeyao than in Purlovia. The jugal also seems to extend more anteriorly in Purlovia (Ivakhnenko, 2011: Fig. 8), but this could be preservation bias due to the breakage of the maxilla.

This new Laurasian clade forms the sister taxon of Whaitsioidea + Baurioidea (Fig. 9), with these three subclades forming an unnamed larger clade (labeled X in Fig. 9). Clade X is supported by: the suborbital bar with slightly lateral expansion (12), an extremely slender postorbital bar (16), the anterior extent of the jugal restricted to the anterior margin of the orbit (37), and the posterodorsal terminal margin of the coronoid process rounded (80). The clade Whaitsioidea + Bauroidea is supported by: the dentary anterior portion continuously tapers to a narrow anterior edge (75), and long and slender humerus (109).

As shown in previous studies, the earliest-diverging members of Akidnognathidae came from Laurasia (Russia and China), whereas the basal members of Whaitsioidea and Baurioidea are from South Africa (Huttenlocker & Smith, 2017; Liu & Abdala, 2017a; Liu & Abdala, 2017b). The current analysis indicates that the clade Whaitsioidea + Baurioidea also had a Laurasian origin. Furthermore, the basalmost eutherocephalian is Scylacosuchus from Russia, supporting a Laurasian origin for Eutherocephalia as well.

The record of therocephalians is largely Gondwanic, and is numerically dominated by the South African record that contains nearly half of the currently valid species. Besides therocephalian species known in the South African Karoo record, three additional endemic species are known from the Permian of East Africa (Tanzania and Zambia; note also that some broadly-distributed taxa like Theriognathus are known from both South and East Africa). Laurasian therocephalians are only known in Russia and China. The majority of therocephalians from Russia are Permian in age, whereas the record from China gives a clearer picture of Triassic diversity. Twelve Permian Russian therocephalian species are currently recognized, in faunas ranging from the Guadalupian to the terminal Lopingian, and represent most of the major therocephalian subclades. By contrast, only five therocephalian species are known in the Russian Triassic (Ivakhnenko, 2011), all of which are bauriamorph baurioids. In China, Caodeyao liuyufengi is the fourth known species from the late Permian, whereas the Chinese Triassic record is currently more prolific with seven species, all of them (with the possible exception of Yikezhaogia megafenestrala) documenting different evolutionary stages of the Baurioidea.

Though therocephalians have a limited record in Laurasia, recent research efforts have resulted in steadily growing knowledge of the Therocephalia in China, hinting that the evolution of this group in the Northern Hemisphere was more complex than previously perceived. The discovery of C. liuyufengi in China adds a new Laurasian-restricted clade to Therocephalia (composed of Caodeyao + Purlovia), indicating that substantial diversification was happening even at higher taxonomic levels in this region.

Conclusion

A new, peculiar therocephalian taxon Caodeyao liuyufengi is described from the upper Permian Naobaogou Formation of China. The new, medium-sized species (∼14 cm of skull length) features a short, high snout and a wide temporal opening with the coronoid process of the mandible separated by a wide space medially to the zygomatic arch. Caodeyao liuyufengi forms a monophyletic group with the Russian Purlovia maxima, making this the only late Permian therocephalian clade represented exclusively in Laurasia.

Recent discoveries in the Naobaogou Formation are providing an enhanced view of therocephalian records in the Chinese late Permian. The known Chinese record is comparatively poor in relation to that from the late Permian of Russia, but our recent contributions indicate the great potential of Chinese late Permian sediments to provide new information on Laurasian tetrapod assemblages. Among these are clues to understanding the evolutionary history of therocephalians in Laurasia (as in the new Caodeyao + Purlovia clade) and even globally (with the recognition of basal akidnognathids such as Shiguaignathus and Jiufengia that indicate a Northern Hemisphere origin for this cosmopolitan clade).

Supplemental Information

Supplemental Information 1 Data matrix

Run the matrix with TNT 1.5 (Goloboff & Catalano, 2016)

Click here for additional data file.

We thank the field team that worked at Daqingshan in 2010 (Jia Zhen-Yan, Li Lu, and Xu Xu). We thank Fu Hua-Lin for fossil preparation, Gao Wei for photographs. We thank Christian Kammerer, Fabien Knoll, Heidi Fourie, and Luisa Pusch for their comments.

Institutional Abbreviations

IVPP Institute of Vertebrate Paleontology and Paleoanthropology, Chinese Academy of Sciences, Beijing, China

PIN Paleontological Institute, Russian Academy of Sciences, Moscow, Russian

RC Rubidge collection, Wellwood, Graaff-Reinet, South Africa

Additional Information and Declarations

Competing Interests

Author Contributions

Data Availability

New Species Registration

The authors declare there are no competing interests.

Jun Liu conceived and designed the experiments, performed the experiments, analyzed the data, prepared figures and/or tables, authored or reviewed drafts of the paper, and approved the final draft.

Fernando Abdala performed the experiments, analyzed the data, prepared figures and/or tables, authored or reviewed drafts of the paper, and approved the final draft.

The following information was supplied regarding data availability:

Holotype of Caodeyao liuyufengi, stored at Institute of Vertebrate Paleontology and Paleoanthropology, Chinese Academy of Sciences, Beijing, China as IVPP V 23298; Holotype of Purlovia maxima, stored at Paleontological Institute, Russian Academy of Sciencies, Moscow, Russian as PIN 1538/47; Holotype of Nanictidops kitchingi, stored at Rubidge collection, Wellwood, Graaff-Reinet, South Africa as RC 49.

The data matrix is available in the Supplemental Files. It is used in the phylogenetic analysis.

The following information was supplied regarding the registration of a newly described species:

Publication LSID: urn:lsid:zoobank.org:pub:EBA6DC8E-45B6-42C7-B570-BFA9559F2795

Caodeyao LSID: urn:lsid:zoobank.org:act:95E242CF-B66D-42BE-82C9-7C40F953A94D

liuyufengi LSID: urn:lsid:zoobank.org:act:8D359DE0-4A49-4BBE-BC7D-B0796555920C.

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
