# Peer review of "The tetrapod fauna of the upper Permian Naobaogou Formation of China: 5. Caodeyao liuyufengi gen. et sp. nov., a new peculiar therocephalian"

_PeerJ, doi:10.7717/peerj.9160_

## Round 0.1 · original submission · Minor Revisions

Heavy copyediting of the text and some editing of the figures are necessary. I also recommend that you amend the specific epithet of your new taxon in order for it to agree with ICZN Art. 31.1.2. (=liuyufengi instead of liuyufengia).

Please, together with your unmarked revised manuscript, provide a marked-up copy as well as a document explaining how you have addressed each of the points raised by the reviewers.

·

Basic reporting

No comment. It is the first report of a new species.
There is a couple of grammatical errors and statements that are not clear.
One Figure either has the wrong caption or wrongly referred to, not sure (Figure 4).

Experimental design

No comment

Validity of the findings

Could do with a conclusion at the end of the text.
Read the Discussion and see if it can't flow a bit better, take one aspect at a time, it jumps a little bit, for instance lines 325 to 329 describes the snout and palate, but jumps back to the snout in line 328.

Additional comments

Lines 17-19. Maybe split into two lines, it is too long and loses its meaning.
Line 22. The coronoid process is mentioned throughout the paper, please check, in some figures the caption is absent, also just rephrase this line to make it clear what the separated space defines.
Line 39. Members.
Line 43. Never were tested = never been tested.
Line 99. A depression - show this depression on the Figure.
Line 84. After eroded - and as a result of this ...
Line 114. The caption on the Figure and mention in text is not the same.
Line 118. Show the depression on the Figure.
Line 146. After (Fourie 1974) add (Fig 5D.)
Line 148. The subarcuate fossa seems not to be on the Figure or is it obscured from view?
Line 192. The large pterygoid bears the palatal process, transverse processess, and posteriorly the diverging quadrate ramus.
Line 251. Again just check that the coronoid process is on all the applicable Figures.
Line 294. The secure, maybe you men the entire record.
Line 299. is recognised as a therocephalian because of the presence of ...
Line 301. the dentary.....
Line 304. Instead of has a quite ... has quite a....
Line 309. moderately.
Line 351. eight to nine.
Line 355. broken maxilla instead of broken of the maxilla.
Line 357-358. Not clear what is stated or relevance.
Line 364. Should it not be ruled instead of overruled.
Line 367. therocephalians.
Line 369. therocephalian.
Line 370. evince? rather show.
Line 373. by now, maybe state presently, more prolific.
Line 374. in the baurioids - as being baurioid.
Figure 2. Can't see mf. on Fig.
Figure 3. C. - absent. also mf. and Pt. in caption.
Figure 5. avsq, Cr, imc, Sp, tso and jf missing from some Figures. Check caption, avp, ipr and Sq missing and should qrPt not be qrpt? Where is the transverse process and So?
Figure 7. Missing Pl.

·

Basic reporting

Caodeyao liuyufengia is an intriguing taxon, indicative of a diverse therocephalian fauna in China in the late Permian. This contribution is a worthy addition to the recent series of papers on this assemblage. However, some revisions to the manuscript are necessary before it can be published. The manuscript is generally well organized, but the language needs some editing. Some sentences and sections are very oddly-phrased and need to be rewritten, with the Discussion especially needing more work. Notable cases are addressed below in the line-by-line comments.

Make sure that the style of in-text citations is consistent and follows the Instructions for Authors of the journal. You often use different styles, e.g. in line 37 it is “(Liu & Bever 2018)”, and in line 319 it is “(Ivakhnenko, 2011)”, or in line 243 it says “(Ivakhnenko, 2011: figure 8)” compared to line 354: “(Ivakhnenko 2011 figure 8)”.

The structure of the manuscript generally conforms to PeerJ standards, but a Conclusion to your findings is missing and still needs to be written.

An additional subheading (“Cranium”) or subheadings is needed in the Description to parallel the “Mandible” section. It would also be good to change the order of the cranial bones you describe, since your current order is somewhat confusing. The vomer and palatal elements (palatine and pterygoid) should be inserted after the description of the maxilla and circumorbital bones, and before the description of the braincase and occipital elements. In the Discussion section a further subheading is needed to differentiate between the the actual discussion of your results and the phylogenetic analysis section.

The figures look generally good, but need some improvements especially regarding stylistic consistency (see below).

Sincerely,
Luisa Pusch & Christian F. Kammerer
(review undertaken collaboratively)

Comments by line:

24: “Phylogenetic analysis indicates a close relationship of the new Chinese taxon with Purlovia maxima, producing the only monophyletic group in therocephalian phylogeny, represented solely in Laurasia.” I think you mean “the only monophyletic therocephalian group represented solely in Laurasia” (no comma), but even this is incorrect—according to your phylogeny the Ordosiidae (Hazhenia + Ordosiodon) is monophyletic, and this clade is also strictly known from Laurasia.

43: Their positions are also not tested in this paper.
49: “a remarkably therocephalian diversity”—this should be “remarkable”, but I would disagree with the statement entirely. Three therocephalians is not a remarkable diversity, considering that comparable late Permian assemblages like the Cistecephalus Assemblage Zone of South Africa yield much higher therocephalian species richness.
76-77: Weird phrasing, better to say: …“except Purlovia maxima, by a deep and laterally bowed zygomatic arch. It is distinguished from P. maxima by having a tall pterygoid median crest, a wide and flat base of the postorbital bar,”…

79-81: “The following characters might represent autapomorphies of C. liuyufengia: a very large fenestra ovalis and a deltopectoral crest longer than half the length of the humerus.”
85-86: “The snout is deflected”…..“along with a wide crack (Fig.1)”. This is also visible in Figure 4.
87: change “right one” to “right ramus”.
87: “A left humerus was found associated with the cranial material.” Refer to your Figure 6 here.
105-106: “The lacrimo-palatine ridge is nearly vertical and deep (Fig. 3)”. This ridge is also visible in Figure 1.
133-134: change “on the occipital view” to “in occipital view”.
135: change to “paroccipital process”
150: “The canals and fossae of this part are similar to those of the Russian therocephalians Chthonosaurus velocidens and Karenites ornamentatus”…
155: change to “presence of a central process of the prootic”.
157 and 173: misspelling of “post-temporal fenestra”
169: Also refer to Figure 5A here.

188: change “The most anterior portion of this bone” to “The anteriormost portion of the parabasisphenoid”
189: What is bifurcated into two ridges? The anterior portion of the parabasisphenoid or that of the keel?
189: change “continues anteriorly with the keel on pterygoid” to “continue anteriorly onto the pterygoid”.
190: …“and form a concave area between them.” Between what?
192-199: Refer to Figures 1, 3 and 4 here.
199-203: Refer to Figures 1, 3 and 4 here as well.
199: “A small ectopterygoid is located lateral to the vacuity”… Please rephrase, because this statement is repetitive to the previous sentence where you already describe the location of the ectopterygoid lateral to the suborbital vacuity.
206: change “In ventral view palatines border…” to “In ventral view, the palatines border…”
211: “Kotelcephalodon” should be “Kotelcephalon”
212: add comma after “missing”
215-216: “If the vomeronasal organ was housed here, it should have being much smaller than in other therocephalians.” The anterior portion of the vomer housing the vomeronasal organ is broken off in this specimen. In the photograph of the actual specimen there is only one median ridge visible along the dorsal surface of the vomer. It seems the three ridges you described on the dorsal side of the vomer are rather caused by damaging of the bone in that area instead of contributing to the housing of the vomeronasal organ, and there is no evidence that the VNO was smaller in Caodeyao than in other therocephalians.
218-219: refer to your Figure 3 here, where these features are visible.

220-221: refer to Figure 1 here again.
222-229: Also refer to your Figures here, where these characters are visible.
232: change … “portion of the right are occluding to the skull” to “portions of the right ramus are occluded to the skull.”
236: “symphyseal”
238: Pusch et al. (2020) discusses this for Lycosuchus
241: “bauriamorphs” instead of “bauriids” (Bauriidae sensu stricto usually treated as Bauria, Microgomphodon, and Traversodontoides, with ordosiids as a separate family and Nothogomphodon outside of both clades)
246: change to …“ramus, resulting in the lower margin of the dentary being slightly concave in lateral view.”
247: delete “lateral” before “sulcus”
250: change ”forms an angle of 140°” to “is about 140°”, since “the angle…forms an angle” is a confusing and somewhat repetitive statement.
255-256: “The left dentary bears one canine and six postcanines”. This is also visible in the right dentary according to Figure 2.
261: change “transverse flange (Fig.3).” to “transverse flange of the pterygoid (Fig. 3).”
299: change to …“Caodeyao liuyufengia can be recognized as a therocephalian by the presence of a large…”
304: Always spell out genus name at the beginning of a sentence.
305: change “similar to non-eutherocephalians Gorynychus masyutinae…” to “similar to the non-eutherocephalians Gorynychus masyutinae…” (also, on what basis are you considering Kotelcephalon a non-eutherocephalian? It is not included in your phylogenetic analysis.)
306-307: change … “Tatarinov 1999); and quite different”… to …“Tatarinov 1999), but quite different”…
307: How does the vomerine interchoanal process differ from that of other eutherocephalians?
314: change “cynodonts (e.g., Procynosuchus, Thrinaxodon; Brink, 1963; Hopson and Kitching, 2001)” to “cynodonts such as Procynosuchus and Thrinaxodon (Brink, 1963; Hopson and Kitching, 2001)” Cite further references such as Kemp (1979); Philosophical Transaction of the Royal Society of London and Jasinoski et al. (2015); The Anatomical Record, here.
327: change to “This palatal morphology is known in some non-bauriid baurioids”
331: change to “There does not seem to be any justification”…
333: refer to Figure 9 here instead of writing “see below”.
333-334: It would be good to add a new subheading here.
343: change to …“and this clade supported by a paired vomer”…
343: …“this clade is supported by a paired vomer and four lower incisors”. The recovery of a classic “Pristerognathia” (consisting of a monophyletic group of basal therocephalians) seems to be based partly on miscodings. Lycosuchus has only three lower incisors (see also Pusch et al., 2020). This is also visible in the Gorynychus specimen described by Kammerer and Masyutin (2018). According to the authors it is likely that more incisors were present, since they do not occupy the entirety of the symphysial length, but it still casts doubt on the definite count of four for the incisors.
345: change to “contacting or nearly contacting”
348: Not true, see comments on Ordosiidae above.
351: change “eight tonine” to “eight to nine”
352: change …“Caodeyao, absent in Purlovia;”… to …“Caodeyao, which are absent in Purlovia.”…
353: change “the base of postorbital bar” to “and the base of the postorbital bar”
355: change “the broken of the maxilla” to “breakage of the maxilla”
356: Pure speculation.
357: What characters support this position?
359-363: Please rephrase these sentences.
362: should be “Laurasian”
364-365: Rephrase sentence.
366-367: Which species do you mean here? Keep in mind that several taxa of South African therocephalians (e.g., Theriognathus) also have an east African record in addition to the endemic forms.
367: change “Therocephalian” to “Therocephalians”
369: misspelling of “therocephalians”
369-371: Some references would be good to add here.

371: Which taxa do you mean here? Some references would also be helpful.
374: How do you know Yikezhaogia is not a baurioid?
388: delete a (in “((Jia Zhen-Yan, Li Lu, and Xu Xu)”)

Figures: 

There are cases in which you use some abbreviations in the figures you forgot mention in the figure captions or mention an abbreviation in the figure caption you do not use in the figures:

Figure 2: “mf” is listed in the caption, but not used in the figure.

“Sp” is used in the figure, but not mentioned in the caption.

Figure 3: You use the abbreviation “Cr” for the coronoid, but in the Figure it is labelled only with a “C”.
“Pt” and “mf” are used in the Figure, but not mentioned in the caption.
Figure 4: “Bo” and “jf” are used in the figure, but not mentioned in the caption.

Figure 5: Image G is not mentioned in the caption.

“avSq” is mentioned in the caption, but not used in the figure.
“avp” is used in the figure, but not mentioned in the caption.
“Cr”, “crch”, “imc”,”Pt”, “Sp”, “tp”, and “V” are mentioned in the caption, but are not used 
and also not visible in the figure. You use the abbreviation “qrPt” for the quadrate process of the pterygoid in the caption, but in the figure it is labelled as “qrpt”.
“tso” is mentioned in the caption, but is missing in the figure.
Figure 7: “Pl” is listed in the caption, but not used in the figure.

Generally, if you want to highlight non-ossified areas in your drawings in grey such as the foramen magnum, the choana, the fenestra ovalis, the post-temporal fenestra and the jugular foramen, make sure that you do this in all images. In Figure 1 you did not highlight the choana, foramen magnum and post-temporal fenestra. In Figure 3 you did not highlight the choana, and in Figure 4 you forgot to do this for the fenestra ovalis, the jugular foramen and the visible foramen magnum, which is not labelled here. In Figure 5 you forgot to highlight the foramen magnum and the left jugular foramen in B, the fenestra ovalis in E, and the foramen magnum, right jugular foramen and vii in F. In Figure 5D there is a foramen in your drawing visible below the foramen magnum, which is not to visible in the actual specimen. In 5E there is a further non-ossified area visible in the actual specimen where the label line of “adp” points to your drawing, which is not highlighted here.
In Figures 2, 6, 7 and 8 we would suggest to add strokes to all label lines and labels pointing to darker bone areas to make them more visible. 

In Figure 6 the label lines and labels are partly displaced such as for f en, dc, or th and cp.

In Figure 7 images A, B, and C are a bit too large compared to the photograph of the mandible in D. It would look better if you would scale A, B and C smaller and D larger that they have a similar size. It would also be good, if the scale bars would have the same thickness.

In Figure 8 avoid it to use different font sizes for your labels. It looks more consistent and neat, if you use the same font size for all labels in your Figures. Also the label line in A, which is labelled as the snout, should be labelled as the maxilla, since the nasal is also part of the snout.

Experimental design

No comment.

Validity of the findings

There is remarkably little discussion of the novel phylogenetic placement of this taxon and Purlovia, considering that you recover them as a completely new lineage of eutherocephalians. Discussion of the character support for this position is essentially absent (by this I do not mean support for a Caodeyao+Purlovia clade, which is discussed, but why these taxa are recovered outside of Whaitsioidea+Baurioidea). No discussion is present either on potential functional or ecological implications of the aberrant morphology of Caodeyao and Purlovia, which beg some degree of interpretation.

Additional comments

No further comments.

---

## Round 0.2 · Minor Revisions

Reviewer 2 has been kind enough to thoroughly copy-edit your text. Please, take these corrections into account. We can then move rapidly to Acceptance. Many thanks, and stay safe.

·

Basic reporting

No comment.

Experimental design

No comment.

Validity of the findings

No comment.

Additional comments

I approve all the changes in scientific content from the first version. I have made a number of additional minor edits to the tracked manuscript to improve the grammar, however (see attached file). Please incorporate these into the final version, at which point this manuscript will be ready for publication.

---

## Round 0.3 · accepted · Accept

I am pleased to confirm that your paper has been accepted for publication.